# Effects of Somatic, Depression Symptoms, and Sedentary Time on Sleep Quality in Middle-Aged Women with Risk Factors for Cardiovascular Disease

**DOI:** 10.3390/healthcare9101378

**Published:** 2021-10-15

**Authors:** Hyun-Sook Choi, Kyung-Ae Kim

**Affiliations:** College of Nursing, Kyungdong University, Wonju 26495, Korea; bright-choi69@kduniv.ac.kr

**Keywords:** somatization symptoms, depression, sedentary time, sleep quality, middle-aged women, cardiovascular risk factors

## Abstract

Cardiovascular disease (CVD) is the second leading cause of death among Korean women, and its incidence is dramatically elevated in middle-aged women. This study aimed to identify the predictors of sleep quality, a CVD risk factor, in middle-aged women with CVD risk factors to provide foundational data for developing intervention strategies for the prevention of CVD. The subjects, 203 middle-aged women (40–65 years old) with one or more CVD risk factors were selected through convenience sampling and included in this descriptive correlational study. The effects of somatic symptoms, depression symptoms, and sedentary time on sleep quality were examined. CVD-related characteristics were analyzed using descriptive statistics, whereas the mean values of the independent variables were analyzed using t-tests and analysis of variance. Predictors of sleep quality were analyzed using multiple regression analysis. The results showed that sleep quality increased with decreasing somatic symptoms (*β* = −0.36, *p* < 0.001), depression symptom score (*β* = −0.17, *p* = 0.023), and daily sedentary time (*β* = −0.13, *p =* 0.041), and the regression model was significant (*F* = 19.80, *p* < 0.001). Somatic symptoms are the most potent predictors of sleep quality in middle-aged women. Thus, intervention strategies that improve somatic symptoms are crucial for the enhancement of sleep quality, which deteriorates with advancing age.

## 1. Introduction

According to the World Health Organization (WHO), 31% of deaths worldwide are due to a cardiovascular disease (CVD) [1], and one out every five women in the United States dies from CVD [2]. In 2018, cancer had the highest mortality rate (22%) in Korean women, followed by cardio-cerebrovascular disease (20%), which is 3% higher than that reported in Korean men (17%) [3]. These statistics highlight the need for adequate interventions for the prevention of CVD.

CVD risk factors include diabetes mellitus (DM), obesity, hypercholesterolemia, and hypertension [4], which are more common in men than in women before middle adulthood; however, after menopause, the prevalence of these risk factors dramatically increases among women compared to the prevalence among men [5]. The reason for the drastic increase in CVD risk factors after menopause is that estrogen deficiency and elevated adrenaline caused by menopause lead to structural and functional changes in the cardiovascular system; these changes increase visceral fat and systemic inflammation, which induce hypertension, impaired glucose tolerance, and increased abnormal lipid and insulin resistance, which then lead to considerable increase in the development of CVD risk factors [5,6].

Somatic symptoms, such as sweating and hot flashes, which commonly occur in menopause as a result of hormonal changes in middle adulthood, deteriorate sleep quality and decrease sleep duration [7,8,9]. Deterioration of sleep quality and duration further increases the risk for developing cardiometabolic risk factors [10], including obesity, hypertension, type 2 DM, and the risk for coronary artery disease [11]. In addition, these hormonal changes lead to insomnia, obstructive sleep apnea, and restless leg syndrome, which diminish sleep quality and increase cardiovascular mortality in women with sleep apnea [12].

The incidence of depression in relation to insomnia and sleep deprivation is higher among women than among men [12]. Depression and somatic symptoms hinder early detection of CVD symptoms in middle-aged women [13,14,15], thus delaying treatment [16]. Moreover, women with depression may have a more sedentary lifestyle with reduced physical activity [17], which further increases the risk for CVD [18].

Thus, there is a growing emphasis on the need for different CVD prevention approaches for men and women due to sex-specific differences in the epidemiology, pathophysiology, clinical management, and outcomes of CVD [19]. The aim of this study was to identify the predictors of sleep quality, a CVD risk factor, in middle-aged women who have a risk for CVD to provide foundational and valuable data for the development of sex-specific interventions and strategic programs for the prevention of CVD in the future.

## 2. Materials and Methods

### 2.1. Research Design

This was a descriptive correlational study conducted to examine the effects of somatic symptoms, depression, and sedentary time on sleep quality in middle-aged women with CVD risk factors.

### 2.2. Setting and Sample

Middle-aged women aged 40–65 years who live in Seoul or two other large cities in South Korea and have at least one CVD risk factor were selected for this study through convenience sampling. Individuals who met at least one of the following criteria proposed by the American Heart Association [20] were selected: overweight (body mass index (BMI) ≥25 m^2^) and particularly those with central adiposity; diagnosis of hypertension or pharmacological therapy for hypertension, diabetes, and symptoms of CVD; diagnosis of hyperlipidemia or pharmacological therapy for hyperlipidemia; family history of CVD; and minimum physical activity (30 min of moderately intense exercise for <5 days per week).

Patients were excluded if they met the following criteria: previous diagnosis of cardio-cerebrovascular disease (e.g., myocardial infarction, stroke, cerebral hemorrhage); diagnosis of depression, anxiety, mental disorder, or cognitive disorder, as determined by a psychiatrist; history of hypothyroidism, previous sinus surgery, or autoimmune diseases; and night shift workers, as this work pattern can affect the quality of sleep.

### 2.3. Sample Size

The sample size of the cohort was determined using a general power analysis program [G*power 3.1, Heinreich-Heine-Universität, Düsseldorf, Germany]. For multiple regression with a medium effect size (f2) of 0.15, power (1-β) of 0.80, significance of 0.05, and eight independent variables (factors associated with the risk for CVD in previous studies (age, hypertension, DM, BMI, sleep duration) and the parameters of the present study (somatic symptoms, depression, sedentary time), the minimum sample size was calculated to be 109. Considering a 20% withdrawal rate, 147 participants were recruited. A total of 203 participants were recruited; thus, the minimum sample size was met.

### 2.4. Data Collection

Data were collected from middle-aged women who visited a private clinic in Seoul, Pyeongtaek-si, Cheong Ju, or Guri-si, or from those working for a production company (number of employees ≥500) between May 2017 and December 2018. The data were collected with the cooperation of a health nurse and the nurse in charge at the clinic. One of the authors and one CVD nurse explained the purpose of the study to the participants, distributed the open-ended and closed-ended questionnaires, and retrieved them immediately after they were completed. Anthropometric measurements were taken after the questionnaires were completed.

### 2.5. General and Anthropometric Characteristics of the Participants

The education, occupation, medical history, current illness, family history, and perceived health of the participants were surveyed. Height and weight measurements were recorded based on self-reported measurements taken in the past month or the measurements the participants recalled. These measurements were used to calculate BMI. To calculate the waist-to-hip ratio (WHR), waist circumference was first measured by gently resting a tape measure (82203-rondo, Korea) around the midpoint between the lowest rib and upper iliac crest while the participant stood upright on a flat surface with legs 25–30 cm apart after breathing out comfortably. Hip circumference was measured by gently pulling the tape measure around the most protruding part of the hip (method proposed by the WHO). The measurements were taken by one of the authors and one nurse using a tape measure from the same manufacturer. The WHR was then calculated using these measurements. The cutoff for WHR was 0.8 or higher [21]

Sleep duration, which was defined as the average sleep duration per week including naps, was evaluated. The participants were also asked how much time they spent per day being sedentary, excluding sleep time.

#### 2.5.1. Quality of Sleep

The Korean Version of the Modified Leeds Sleep Evaluation Questionnaire was used to evaluate sleep quality. This 10-item tool evaluates getting to sleep, awakening from sleep, perceived quality of sleep, and behavior after waking. Each item is rated on a 10-point scale, and the total score ranges from 0 to 100. The optimal cutoff point is 67; a score of 67 or lower indicates poor sleep quality, whereas a score of 68 or higher indicates good sleep quality [22]. The Cronbach’s alpha for this tool was 0.88 at the time of development [23] and in this study.

#### 2.5.2. Somatic Symptoms

Somatic symptoms were measured using the Symptom Checklist-90-Revision-Somatization (SCL-90R-SOM) (Central Aptitude Publishing Department, 1984, Korea) tool derived from the Korean version of the Mini-Mental State Exam. The SCL-90R-SOM comprises 12 items about somatization symptoms, which are rated on a four-point Likert scale. The total score ranges from 0 to 48, with a higher score indicating more severe somatization symptoms [24]. The reliability of the tool (Cronbach’s alpha) was 0.93 at the time of development [25] and 0.88 in this study.

#### 2.5.3. Depression Symptoms

The Patient Health Questionnaire-9 was used to evaluate depression. This tool assesses the frequency of symptoms described in each item over the past two weeks. The questionnaire contains nine items rated on a four-point scale from 0 (never) to 3 (almost every day). The total score ranges from 0 to 27, with a score of 10 or higher indicating clinical depression. The reliability of the tool (Cronbach’s alpha) was 0.95 at the time of development [24,25,26] and 0.82 in this study.

### 2.6. Ethical Considerations

The study protocol was approved by the Institutional Review Boards of Hanyang University (HYI-14-118-3) and Kyungdong University (1041455-202012-HR-010-01). The participants were informed that all personally identifiable information will be removed from the data used for analysis to protect their identity. Written informed consent was obtained all subjects involved in the study. The collected data were placed in a locked cabinet that could only be accessed by the researchers. The participants were informed that the data would be shredded for disposal upon completion of the study. The anthropometric measurements were taken in a relaxed environment in a counseling room to ensure privacy. Participants who completed the survey were given a small gift.

### 2.7. Statistical Analysis

The collated data were analyzed using SPSS 21.0 (IBM SPSS Statistics, Chicago, IL, USA). The general and CVD-related characteristics of the participants were analyzed using descriptive statistics, whereas the differences in sleep quality according to the independent variables (depression and somatic symptoms) were analyzed using t-tests and analysis of variance. Correlations among sleep quality, general characteristics, and independent variables were analyzed using Pearson’s correlation coefficient. Prior to the identification of the predictors of sleep quality, multicollinearity among the independent variables was tested in step 1 using the tolerance and variance inflation factor (VIF). In step 2, multiple linear regression was performed to identify the independent predictors of sleep quality and to examine the percentage of variance explained by each factor. The reliability of the tools used in this study was evaluated using Cronbach’s α. Statistical significance was set at *p* < 0.05.

## 3. Results

### 3.1. General Characteristics of the Participants

The mean age of the good and poor sleep quality groups was 55.74 ± 6.33 years and 53.67 ± 6.93 years, respectively. The mean ages of the two groups were significantly different (*t* = −2.21, *p* = 0.028). Regarding occupation, 29 (27.4%) and 21 (22.1%) participants in the poor sleep quality group were manual laborers and service workers, respectively, whereas 63 (52.7%) participants in the good sleep quality group were manual laborers (*x*^2^ = 14.52, *p* = 0.006). Type of occupation significantly differed between the two groups. However, perceived health status was not significantly different between the two groups (*F* = 0.73, *p* = 0.693) (Table 1).

### 3.2. Cardiovascular Disease-Related Characteristics of the Participants

CVD risk factors and related variables and smoking were significantly associated with sleep quality (*x*^2^ = 6.91, *p* = 0.032). Daily average sedentary time was 5.21 ± 3.14 in the good sleep quality group and 7.24 ± 8.16 in the poor sleep quality group. Sedentary time was significantly different between the two groups (*t* = 2.33, *p* = 0.021). Sleep duration also significantly differed between the two groups (*x*^2^ = 5.35, *p* = 0.021); 44.7% of participants in the good sleep quality group and 28.9% of those in the poor sleep quality group reported an adequate sleep duration of 7–8 h. Depression symptoms were significantly higher in the poor sleep quality group than in the good sleep quality group (*t* = 5.35, *p* < 0.001). The somatic symptom score was also significantly higher in the poor sleep quality group than in the good sleep quality group (*t* = 4.19, *p* < 0.001). There was no significant difference in the age at menopause and the presence or absence of menopause between the two groups (*t* = −0.19, *p* = 0.848) (Table 2).

### 3.3. Correlation between Sleep Quality and the Independent Variables

Analysis of the correlation between sleep quality and the independent variables showed that sleep quality was negatively correlated with somatic symptoms (*r* = −0.47, *p* < 0.001), depression symptoms (*r* = −0.39, *p* < 0.001), and daily average sedentary time (*r* = −0.15, *p* = 0.041). Depression was positively correlated with somatic symptoms (*r* = 0.57, *p* < 0.001) (Table 3).

### 3.4. Comparison of Somatic Symptoms in the Two Groups

Except for foreign body sensation in the throat, all somatic symptoms significantly differed between the two groups (*p*-value, 0.000–0.031). The most common symptom was low back pain, with a score of 1.54 in the good sleep quality group and 1.80 in the poor sleep quality group. The second most common symptom was heaviness in the legs, with a score of 1.42 in the good sleep quality group and 1.42 in the poor sleep quality group. Having a foreign body sensation in the throat was the least common symptom in both groups, with no significant difference between the two groups (*p* = 0.088) (Figure 1).

### 3.5. Predictors of Sleep Quality in Middle-Aged Women with CVD Risk Factors

To test the assumptions of the regression, the autocorrelation (independence) of the independent variables was tested using the Durbin–Watson statistic. The results confirmed the absence of autocorrelation. Regarding multicollinearity, tolerance (0.75–0.98) was below 1.0, whereas VIF (1.01–1.33) was below 10, confirming the absence of multicollinearity. Regression analysis was performed to confirm whether somatic symptoms, depression, and sedentary time, which were significant in the univariate analysis, were independent predictors of sleep quality. The regression model was significant, with the independent variables explaining 24% of the variance (*F* = 19.80, *p* < 0.001). In this model, sleep quality increased with decreasing somatic symptom score (*β* = −0.36, *p* < 0.001), depression symptom score (*β* = −0.17, *p* = 0.023), and daily average sedentary time (*β* = −0.13, *p* = 0.041) (Table 4).

## 4. Discussion

This study was conducted to identify the predictors of sleep quality that increase the risk for CVD in middle-aged women with pre-existing CVD risk factors. Our results showed that somatic symptoms are the most potent predictors of sleep quality. This finding is consistent with the results of a cohort study of 10,000 participants aged 35–74 years old who were followed up for five years from 2007 to 2012. The results of that study indicated that somatic symptoms and pain are predictors of sleep quality in both men and women [27]. Furthermore, a study of 236 Hong Kong families (224 mothers and 196 fathers; mean age, 47 years old) demonstrated that somatic symptoms and pain are important predictors of sleep quality, a result which supports our findings [28]. Pain, repeatedly identified as a predictor of sleep quality, is also a somatic symptom. In the present study, numbness in the legs and heaviness or pain in the extremities were significantly more common in the poor sleep quality group than in the good sleep quality group. This result also supports the findings of previous studies. Somatic symptoms were positively correlated with depression and sedentary time in the present study. This is consistent with findings of a previous cross-sectional study of 960 Korean adults aged 45 years or older, in which somatic symptoms were positively correlated with depression symptoms and negatively correlated with physical activity [29].

In the present study, depression symptoms were the second most potent predictors of sleep quality. This finding is similar to that of a study of 817 elderly people, in which the prevalence of depression increased with deteriorating sleep quality [30].

It is well known that sleep quality ultimately influences sleep duration [31]. In the present study, 77.1% of participants in the poor sleep quality group and 55.3% of those in the good sleep quality group had an inappropriate sleep duration (<7 h or >8 h). This is consistent with the results of a 2012–2013 survey of 11,276 adults in Northeast China aged ≥35 years old that demonstrated increased prevalence of depression among those who had too short (≤6 h) or too long (≥9 h) sleep duration [32].

Notably, women experience depression more frequently than men [19]. A four-year follow-up study of 93,676 postmenopausal women showed that depression increases the incidence of CVD even after adjusting the general risk factors of CVD [33]. The results of these previous studies and those of the present study suggest that depression must not be neglected and must be appropriately managed to reduce the occurrence of somatic symptoms. According to the “Fasa PERSIAN Cohort Study” of 10,129 Iranian subjects aged 35–70 years, a shorter sleep duration (≤6 h) was associated with a 1.2-fold increased incidence of CVD. This finding was consistent with that of the Framingham risk score for short sleepers [34]. In this study, the majority (61%, 126 participants) of middle-aged women with CVD risk factors had poor sleep duration (<7 or >8 h). This emphasizes the need for interventions to improve women’s sleep quality.

Depression has been reported to be positively correlated with sedentary time [17,29,35]. Our findings showed that sedentary time and depression are positively correlated. We also found that increased sedentary time contributes to deterioration of sleep quality. This is consistent with the findings of the Sleep in America poll of 843 adults, in which sleep quality was found to decline with increasing screen time (i.e., television viewing and computer use during leisure time) [36]. An increase in sedentary time not only affects sleep quality, but also increases the incidence of chronic disease and CVD [36,37,38]. Thus, aggressive interventions that promote physical activity in middle-aged adults are necessary to prevent and reduce CVD morbidity. In addition, improving sedentary lifestyle habits and increasing physical activity is important for the enhancement of sleep quality, which has been found to reduce depression, somatic symptoms, and CVD risk factors in women [39,40,41]. However, women generally engage in less physical activity and demonstrate low compliance compared to men [42]. In a study of 846 individuals with CVD risk factors and normal individuals, the percentage of individuals who met the recommended 600–1500 MET of physical activity was 28% in the CVD risk group and 31% in the control group, indicating that people with CVD risk factors engage in less physical activity than healthy individuals [43]. Thus, physical activity must be promoted among women to prevent CVD.

Although we did not directly measure the amount of physical activity among our participants, the poor sleep quality group had markedly higher sedentary time than the good sleep quality group. Increased sedentary time further elevates the risk for CVD in people with poor sleep quality. Thus, sex-specific intervention programs that facilitate physical activity are needed to enhance sleep quality and lower CVD morbidity among individuals with CVD risk factors. However, a prior study revealed that work-related sedentary time is not a consistent risk factor for CVD [44]. This disparity in the findings of previous studies indicates that further research is needed to clarify the association between sleep quality and the incidence of CVD according to sedentary time in women.

This study has a few limitations. First, this study was focused on understanding the relationships among the variables, and the participants were selected using convenience sampling. Second, patients with sinus hypertrophy, obstructive sleep apnea, and narcolepsy, which directly influence sleep quality, could not be excluded in advance. Hence, the results have limited generalizability. Thus, future multicenter studies are needed to corroborate the findings of this study. Finally, this study is a descriptive survey; therefore, causality among the variables could not be established. Thus, intervention-based longitudinal studies with long follow-up periods focused on the enhancement of sleep quality in women are needed in the future.

Nevertheless, this study is significant in that it identified the predictors of sleep quality, a CVD risk factor, and shed light on the relationship between somatic symptoms, depression, and sleep quality in middle-aged women with CVD risk factors. Hence, the results provide valuable foundational data for the development of customized intervention programs and sex-specific strategic education programs aimed at reducing CVD risk factors in women.

## 5. Conclusions

This study was conducted to identify the predictors of sleep quality, a CVD risk factor, in middle-aged women with CVD risk factors. Somatic symptoms, depression symptoms, and sedentary lifestyle were identified as predictors of sleep quality. The results also showed that sleep quality deteriorates with an increase in each variable. The most potent predictor of sleep quality was somatic symptoms. These findings suggest that implementing sex-specific interventions and lifestyle modifications could be effective in reducing the development of certain CVD risk factors. Moreover, the findings provide useful data that can facilitate the planning of intervention strategies to ameliorate the deterioration of sleep quality resulting from hormonal changes in middle-aged women.

## Figures and Tables

**Figure 1 healthcare-09-01378-f001:**
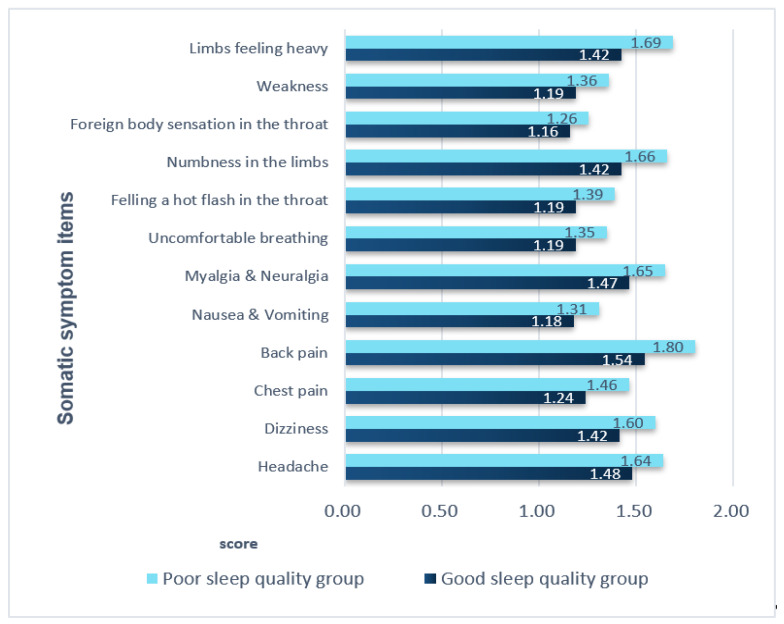
Comparison of somatic symptoms in the two groups. All variables except for the variable of foreign body sensation in the throat had *p*-values < 0.05 (range: 0.000–0.031). For the variable ‘foreign body sensation in the throat’, the *p*-value was *p* = 0.088.

**Table 1 healthcare-09-01378-t001:** General characteristics of the participants. (*N* = 204).

Variables	Categories	Good Sleep Quality Group (*n* = 106)	Poor Sleep Quality Group (*n* = 97)	t or *x*^2^	*p*
		*n* (%) or Mean ± SD	*n* (%) or Mean ± SD		
Age (yr)		55.74 ± 6.33	53.67 ± 6.93	−2.21	0.028
	40∼50 51~60 61~65	27(25.7) 50(47.6) 28(26.7)	36(27.0) 43(44.4) 18(18.6)	6.06	0.195
Menopause age		50.92 ± 4.10	50.79 ± 3.09	−0.20	0.843
Presence of menopause	Yes No	75(71.4) 30(28.6)	59(62.8) 35(37.2)	1.69	0.193
Education level	≤Middle school High school ≥College	38(36.2) 52(49.5) 15(14.3)	27(28.1) 47(49.0) 22(22.9)	3.04	0.219
Living with	Spouse or children Alone Living with someone outside the family	83(79.0) 44(10.5) 11(11.0)	77(79.4) 10(10.3) 10(10.4)	1.15	0.765
Occupation	Managerial/Official Service/Sales Professional Laborer Unemployed/Housewife	7(6.6) 7(6.6) 12(11.3) 63(52.7) 17(16.0)	11(11.6) 21(22.1) 9(9.5) 29(27.4) 17(19.9)	14.52	0.006
Perceived health status	Good Moderate Poor	20(18.7) 63(60.0) 22(21.0)	16(16.5) 56(57.7) 25(25.8)	0.73	0.693

Abbreviation: SD, standard deviation.

**Table 2 healthcare-09-01378-t002:** Cardiovascular disease-related characteristics of the participants. (*N* = 204).

Variables	Categories	Good Sleep Quality Group (*n* = 106)	Poor Sleep Quality Group (*n* = 97)	t or *x*^2^	*p*
		*n* (%) or Mean ± SD	*n* (%) or Mean ± SD		
Height (cm)		156.99 ± 5.31	157.70 ± 5.18	0.97	0.336
Weight (cm)		61.00 ± 8.98	60.49 ± 8.01	−0.43	0.671
Waist (cm)		84.89 ± 9.10	83.54 ± 9.31	−1.04	0.299
Hip (cm)		104.69 ± 84.67	95.47 ± 8.32	−1.12	0.299
Waist-to-hip ratio		0.87 ± 0.10	0.87 ± 0.09	−1.02	0.308
	<0.8 ≥0.8	14(13.2) 92(86.8)	6(6.0) 91(93.0)	0.48	0.634
Body mass index (m^2^/kg)		27.74 ± 3.33	24.30 ± 2.78	−1.02	0.308
	<25 ≥25	61(58.0) 45(42.0)	67(69.1) 30(30.9)	3.77	0.152
CVD risk factors ^†^	Hypertension Diabetes Hyperlipidemia Angina and arrhythmia symptoms ^∫^ Chronic kidney disease Arthritis Fatty liver	60(74.1) 21(25.9) 34(42.0) 5(6.1) 0(0.0) 4(4.1) 3(3.7)	54(68.4) 10(12.7) 44(55.7) 3(3.8) 2(2.5) 11(13.9) 6(7.6)	0.20 3.54 4.51 0.55 0.29 7.95 0.25	0.990 0.060 0.105 0.723 0.557 0.019 0.298
CVD risk factors related to family history ^‡^	Hypertension Diabetes Stroke, MI Cancer	47(44.3) 31(29.2) 21(19.8) 23(21.7)	45(46.4) 31(32.0) 20(20.6) 27(27.8)	0.09 0.18 1.13 1.03	0.769 0.675 0.568 0.311
Smoking	Never Ex-smoker Current smoker	95(91.3) 8(7.7) 1(1.0)	86(89.6) 3(3.1) 7(7.3)	6.91	0.032
Alcohol	<1/month 2~3/months ≥2~4/weeks	71(71.9) 18(17.3) 15(14.4)	61(64.2) 26(27.4) 8(8.4)	5.33	0.256
Menopause (yr)	Yes No Estrogen therapy (Yes)	50.92 ± 4.10 75(71.4) 30(28.6) 0(0.0)	50.79 ± 3.09 59(62.8) 35(37.2) 0(0.0)	−0.19 0.19	0.848 0.227
Sedentary time (1 day)		5.21 ± 3.14	7.24 ± 8.16	2.33	0.021
Sleeping hours (1 day)	Poor (<7 or >8) Good (7~8)	57(55.3) 46(44.7)	69(71.1) 28(28.9)	5.35	0.021
Quality of sleep (score)		79.97 ± 8.53	53.52 ± 10.76	−19.29	<0.001
Depression symptoms (score)		3.76 ± 4.16	6.32 ± 4.89	4.00	<0.001
	No (<10) Yes (≥10)	96(90.6) 10(9.4)	75(77.3) 22(22.7)	6.70	0.010
Somatic symptoms (score)		5.79 ± 5.46	9.72 ± 7.72	4.19	<0.001

Abbreviation: SD, standard deviation; CVD, cardiovascular disease; MI, myocardial infarction. ^†^ Multiple responses, classification % in sleep quality; ^‡^ multiple responses; angina and arrhythmia symptoms, ^∫^ = participants with symptoms of angina and arrhythmia without a formal diagnosis and not taking medications.

**Table 3 healthcare-09-01378-t003:** Correlation between sleep quality and the independent variables (*N* = 204).

Variables.	(1)	(2)	(3)	(4)
	R(*P*)			
(1) Quality of sleep (score)	1			
(2) Somatic symptoms (score)	−0.47 **(<0.001)	1		
(3) Depression symptoms (score)	−0.39 **(<0.001)	0.57 **(<0.001)	1	
(4) Sedentary time (1 day)	−0.15 *(0.041)	0.00(0.990)	0.09(0.196)	1

Correlation coefficients and *p*-values of the nominal items. * Correlation is significant at 0.05 (two-tailed analysis). ** Correlation is significant at 0.01 (two-tailed analysis).

**Table 4 healthcare-09-01378-t004:** Predictors of sleep quality in middle-aged women with CVD risk factors (*N* = 204).

Variables.	B	SE	β	t	*p*
(Constant)	79.40	1.97		40.28	<0.001
Somatic symptoms (score)	−0.92	0.19	−0.36	−4.65	<0.001
Depression symptoms (score)	−0.61	0.27	−0.17	−2.30	0.023
Sedentary time (1 day)	−0.35	0.17	−0.13	−2.06	0.041
*R*^2^ = 0.24, Adjusted *R*^2^ = 0.23, *F* = 19.80, *p* < 0.001					

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
