# Peer review of "Effects of Somatic, Depression Symptoms, and Sedentary Time on Sleep Quality in Middle-Aged Women with Risk Factors for Cardiovascular Disease"

_healthcare, 2021, doi:10.3390/healthcare9101378_

Round 1

Reviewer 1 Report

Overview: In this manuscript titled, "Effects of somatic symptoms, depression and sedentary ttoeme on sleep quality in middle aged women with risk factors for cardiovascular disease" Choi and Kim et. al attempt to describe the predictors of sleep quality in middle-aged women. This research question is novel, timely and relevant. The analysis suggests that somatic symptoms are the most potent predictors of sleep quality in middle-aged women. Prior to consideration for publication, the following concerns need to be addressed.

1.     Introduction: “CVD is the second leading cause of death among Korean Women…” Authors should rephrase to include the leading cause of death and then the CVD and include percentages so show the severity on the CVD and its impact on death.

  1. In the setting and sample section located in the materials/methods section, the authors excluded patients with cardio-cerebrovascular disease and mental disorder but included patients with angina, “etc” and depression as located in table 2. Please clarify precise definitions for inclusion and exclusion criteria and remove “etc” to define what you are referring to.
  2. There is no mention of sleep disorders (obstructive or central or mixed sleep apnea, narcolepsy) in the inclusion or exclusion criteria as these disorders can extensively impact sleep quality. Any major surgeries (eg sinus surgery, etc) that can impact sleep quality? Please re-assess as this is a major limitation if not addressed properly in the manuscript.
  3. Likewise there is no mention of other medical disorders such as hypothyroidism or autoimmune conditions that can impact sleep quality. Please re-assess.
  4. In the data collection section, why were participants from a “food production company”. Is there any bias or disclosures that need to be disclosed? Were participants paid for their enrollment in the study?
  5. The authors state that obesity is a RF for CVD however it seems that they enrolled only patients whom are overweight. No reported patients had a BMI > 30 in the manuscript. Please address to raise the point that even being overweight and not necessarily obese may be a RF.
  6. With all of the co-morbid conditions collected, were any patients on medications that can impact their sleep (eg steroids)? Were any patients on estrogren therapy for menopause? Were any patients on anti-depressants or anti anxiety medications?
  7. Were any workers working at night predominantly and not during the day, which can impact sleep quality?

Reviewer 2 Report

Accept after minor revision (corrections to minor methodological errors and text editing)

Reviewer 3 Report

The authors should be commended for their work that went into preparation of this interesting manuscript.  The Abstract is clear and concise.  The methodology gives a detailed description of all variables and how each one was measured or calculated.

I would like to know from the authors, would the cardiovascular risk profile of their participants differed significantly if they had recruited their sample from public hospitals and how would this have affected the identification of predictors of sleep quality and eventually on the conclusion of their study?

In the discussion, the authors made reference to multiple studies done on similar research.  However, they do not comment on the conflicting results in the literature where some studies show long hours of sleep increase cardiovascular risk, while others state short hours of sleep as the culprit.

I also appreciate your statement on limitations of the study, where the generalizability of the study is acknowledged 

Round 2

Reviewer 1 Report

Thank you for your thoughtful revisions. No further clarifications are needed.

Author Response

We would like to thank you and the reviewers for your insightful comments and helpful suggestions, which have helped us significantly improve the quality of our work. 

Thank you for your consideration. 

KyungAe Kim